# Fungus-originated glucanase and monooxygenase genes in creeping bent grass (*Agrostis stolonifera* L.)

**Yugo Watanabe**[1], **German C. Spangenberg**[1,2], **Hiroshi Shinozuka**[1] *

1 Agriculture Victoria, AgriBio, Centre for AgriBioscience, La Trobe University, Bundoora, Victoria, Australia,
2 School of Applied Systems Biology, La Trobe University, Bundoora, Victoria, Australia

* hiroshi.shinozuka@agriculture.vic.gov.au

**Data Availability Statement:** All relevant data are within the manuscript and its Supporting Information files.

## Abstract

Recent studies have revealed presence of fungus-originated genes in genomes of cool-season grasses, suggesting occurrence of multiple ancestral gene transfer events between the two distant lineages. The current article describes identification of glucanase-like and monooxygenase-like genes from creeping bent grass, as lateral gene transfer candidates. An *in silico* analysis suggested presence of the glucanase-like gene in *Agrostis*, *Deyeuxia*, and *Polypogon* genera, but not in other species belonging to the clade 1 of the Poeae tribe. Similarly, the monooxygenase-like gene was confined to *Agrostis* and *Deyeuxia* genera. A consistent result was obtained from PCR-based screening. The glucanase-like gene was revealed to be ubiquitously expressed in young seedlings of creeping bent grass. Although expression of the monooxygenase-like gene was suggested in plant tissues, the levels were considerably lower than those of the glucanase-like gene. A phylogenetic analysis revealed close relationships of the two genes between the corresponding genes in fungal endophyte species of the *Epichloë* genus, suggesting that the genes originated from the *Epichloë* lineage.

## Background

The genus *Agrostis* [Agrostidinae subtribe, Poeae tribe clade 1 (PC1), Pooideae sub-family (cool-season grasses)] includes around 200 species, and its representative species, such that creeping bent grass and common bent (*Agrostis capillaris*), are grown world-wide as turf [1]. *Deyeuxia angustifolia* [(Komarov) Y. L Chang] and beard-grass (*Polypogon fugax* Nees ex Steud.) also belong to the Agrostidinae subtribe, which are known as native and invasive plants [2–4]. Into the same clade of the Poeae tribe, oat (*Avena sativa* L.) and harding grass (*Phalaris aquatica* L.) are classified, which are cultivated as food and pasture crops, respectively [5].

Fungal endophyte species of the *Epichloë* genus may establish symbiosis with plants of the Pooideae sub-family, providing abiotic and biotic stress resistances to the plant host [6]. *E. amarillans* is known as symbiont of an *Agrostis* species, tickle grass (*A. hyemalis*) [7]. Similarly, plants belong to the genus *Lolium* [Poeae clade 2 (PC2)] may establish symbiotic relationships with *E. festucae*. To date, four genes from cool-season grasses have been identified as

**Funding:** This work was supported by funding from Agriculture Victoria. The funders had no role in study design, data collection and analysis, decision to publish, or preparation of the manuscript.

**Competing interests:** The authors have declared that no competing interests exist.

candidates for laterally transferred genes from the *Epichloë* lineage. The ß-1,6-glucanase hydrolyses ß-1,6-glucans, which are one of main components of fungal cell walls, and the glucanase was, therefore, believed to be confined to mycoparasitic fungi until recent [8]. From perennial ryegrass (*Lolium perenne* L., *Lolium* genus), a ß-1,6-glucanase-like gene, designated *Lp*BGNL, was found, showing a 90% DNA sequence identity to the *E. festucae* ß-1,6-glucanase gene. As putative orthologues of *Lp*BGNL were only found from subtribes Lollinae and Dactylidinae (PC2), but not from other plant lineages investigated, it was suggested that the gene was acquired from the *Epichloë* lineage through horizontal gene transfer (HGT) [9]. Later, a domain of unknown function gene, *Lp*DUF3632, was identified from perennial ryegrass, as HGT candidate. This gene was only conserved in the Loliinae subtribe, and showed a higher DNA sequence identity (96%) to *Epichloë* DUF3632 genes [10]. Another candidate, designated *Fhb7*, was identified from a wheat relative species, *Thinopyrum elongatum* (Triticeae tribe) [11]. The products of this gene had glutathione S-transferase activities and contributed to resistance against fungal pathogens of *Fusarium*. The DNA sequence identity between the *Fhb7* gene and the *Epichloë* counterparts was up to 97%. The fungal transcriptional regulatory protein-like (FTRL) gene was identified to be more widely conserved in the Poeae and Triticeae tribes [10]. The perennial ryegrass FTRL gene, *Lp*FTRL, showed 85% DNA sequence identity between the corresponding *Epichloë* gene. Due to absence in plant lineages other than the two tribes, the FTRL gene was likely to have transferred from the *Epichloë* lineage before diversification of the Poeae and Triticeae tribes.

Horizontally transferred candidates can be identified through a comparison of DNA sequences between two taxonomically distant species [12]. In case of eukaryotes, some constitutive genes, such as actin and ubiquitin genes can be highly conserved even between the two distant taxa, and sequence similarity hits related to those genes need to exclude from the HGT candidates [10]. The following phylogenetic analysis can reveal DNA sequences with an unusually high identity between the two species, to confirm that the gene was transferred across species boundaries. The current study reports the identification of two HGT candidates from creeping bent grass, which show unusually high DNA sequence identities to those of *Epichloë*.

## Materials and methods

### *In silico* analysis

The creeping bent grass transcriptome shotgun assembly (TSA) data (NCBI GenBank: GFQK00000000.1) were obtained from NCBI website, and *E. amarillans* (strain E57/ATCC 200744) transcriptome data (NCBI BioProject: PRJNA67301) were downloaded from the Genome Projects at University of Kentucky website (http://www.endophyte.uky.edu/) [6]. A DNA sequence homology search was performed with the BLAST+ package, using the mega-blast function (https://blast.ncbi.nlm.nih.gov/Blast.cgi). A subsequent manual examination was performed using the NCBI BLAST tool. Alignment of DNA and amino acid sequences was performed using the CLUSTALW program (https://www.genome.jp/tools-bin/clustalw). Phylogenetic analysis was performed using the MEGA X program [13]. An *in silico* screening was performed using the NCBI BLAST tool and short read archive (SRA) database.

### DNA extraction

Plants seeds were obtained from the South Australian Research and Development Institute (SARDI) (S1 Table). The seeds were germinated on filter paper in petri dishes. Total DNA was extracted from young seedings of each genotype using the E.Z.N.A.® Plant DNA Kit (OMEGA).

## PCR-based screening

PCR primers were designed with a support of the OligoCal program (S2 Table). A PCR assay was performed using the Luna Universal qPCR Master Mix kit [New England BioLabs (NEB)] on the CFX Connect Real-Time PCR Detection System (BioRad). As the *Epichloë* genus and its next close genus, *Claviceps*, include species infectious to grass plants of the Pooideae sub-family, absence of those fungal species in plant genomic DNA (gDNA) samples was confirmed with *Epichloë* and *Claviceps*-specific primers [10,14,15].

## PCR-restriction fragment length polymorphism (RFLP) assay

Through an alignment of sequences from creeping bent grass and *Epichloë* species, single nucleotide variations (SNVs) were identified, which could be used for a PCR-RFLP assay. The DNA fragments including the SNV site was amplified with the AsFMOL_con_f1_AstII and r primers, and the amplicons were treated with the *Aat*II restriction enzyme (NEB). The treated amplicons were visualised on a 2% (w/v) agarose gel stained with SYBR Safe (Life Technologies).

## Reverse transcript (RT)-real time PCR (qPCR) assay

Based on morphology, 10 plant genotypes of the creeping bent grass young seedlings were dissected into 3 parts, roots, lower leaves, and upper leaves. Total RNA was extracted from each plant part, using the RNeasy Plant Mini Kit (QIAGEN). cDNA samples were prepared using the Maxima H Minus Reverse Transcriptase kit (Thermo Fisher SCIENTIFIC). The reverse transcript products (20 μl) was diluted with the same volume of 1x TE buffer. A qPCR assay was performed using the Luna Universal qPCR Master Mix kit on the CFX Connect Real-Time PCR Detection System. The creeping bent grass Actin gene, *AsAct* (GeneBank UI: JX644005.1), was used as internal control, and 3 technical replicates were performed for each assay [16]. Relative gene expression levels were calculated using the $2^{-\mu\mu}$Ct method, based on the Cq (threshold cycle) values from the samples. PCR amplification efficiency was examined through a qPCR-based standard curve assay.

## Results and discussion

Using the BLAST+ package, a total of 273 sequence similarity hits between *A. stolonifera* and *E. amarillans* transcriptome were obtained (S3 Table). The subsequent manual examination identified ß-1,6- glucanase-like (GFQK01230513.1) and flavin-containing monooxygenase (FMO)-like (GFQK01114691.1) sequences from creeping bent grass, which were designated *As*BGNL and *As*FMOL. The DNA sequence identity between *As*BGNL and the corresponding *E. amarillans* gene (augustus_masked-contig00145-processed-gene-0.47-mRNA-1) was 93%, and the alignment length (length of homologous region) was 1153 bases, with 79 nucleotide mismatches (S1 Fig). The sequence corresponding to the aryl-phospho-beta-D-glucosidase domain of the glucanase was identified, inferring that the gene products retain the molecular function (Fig 1). Between *As*FMOL and the corresponding *E. amarillans* gene (fgenesh_-masked-contig00237-processed-gene-0.48-mRNA-1), two separated hits were obtained, and the DNA sequence identities were 95.5 and 95.6%. The alignment length for *As*FMOL was 1447 bases, when the two hits were combined (S2 Fig).

A BLAST search against publicly available datasets was performed to examine the presence/absence status of the HGT candidates in taxonomically related plant species. Through the BLAST search against the SRA data of *Agrostis*, *Deyeuxia*, and *Polypogon* species, DNA sequence similarity hits for *As*BGNL were obtained from all tested datasets (Table 1). No

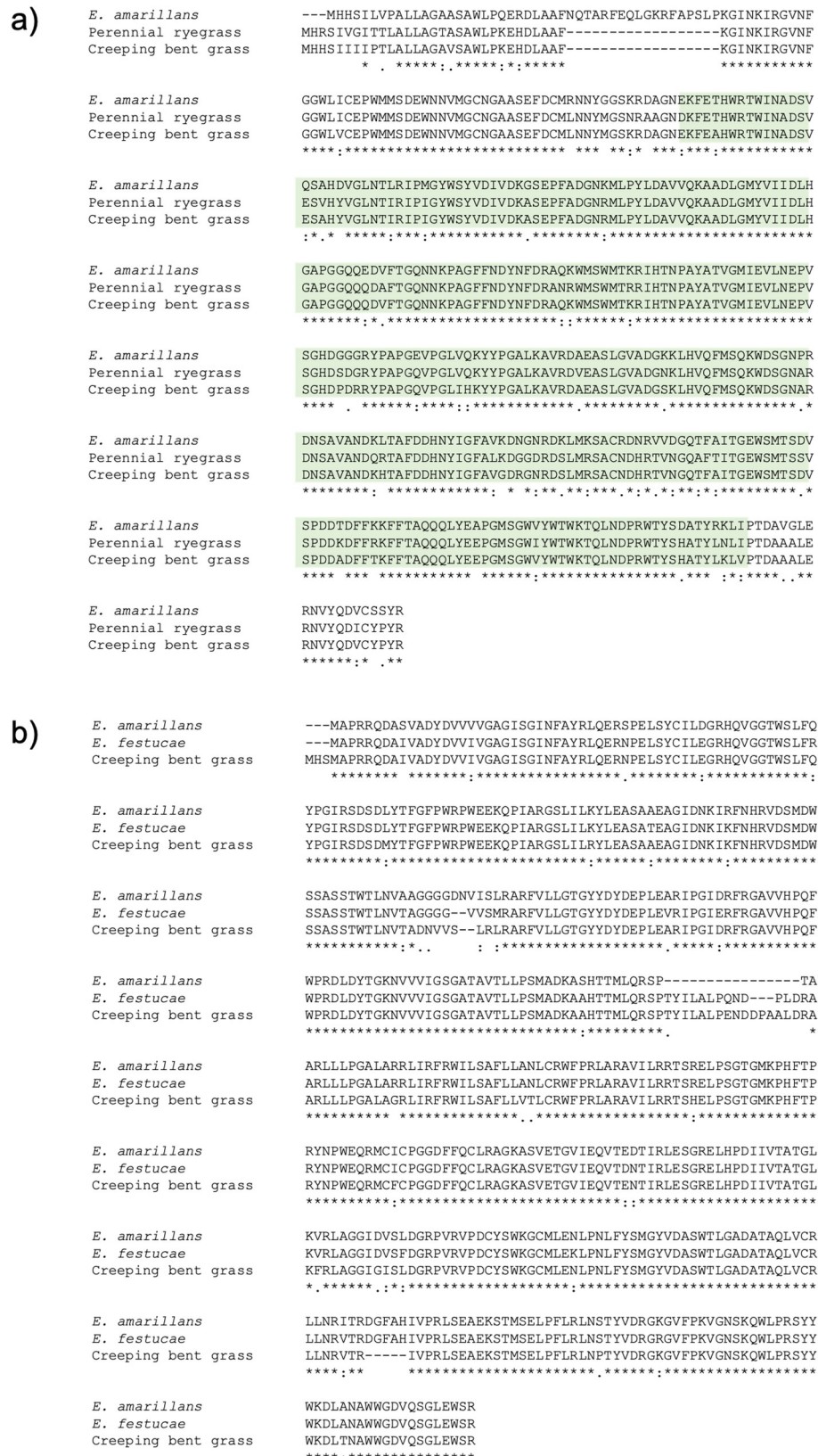

**Fig 1. Amino acid sequence alignments for the predicted products of glucanase-like (a) and flavin-containing monooxygenase-like (b) genes.** The alignments are shown in the CLUSTAL W format. In the amino acid sequences, dash (−) represents a gap. Under the alignments, 'conserved amino acid residues', 'including conserved substitution(s)' and 'including semi-conserved substitution(s)' are denoted with asterisk (*), colon (:) and dot (.). The part of the aryl-phospho-beta-D-glucosidase domain is highlighted with light green (a).

corresponding sequence was, however, found from oat and harding grass. When DNA sequences catalogued in the NCBI GenBank database were sought, the BLAST-based approach revealed a relatively high sequence similarity between *As*FMOL and monooxygenase(-like) genes from fungi of the Clavicipitaceae family (Ascomycota) (S4 Table). Although some partial similarities were observed between sequences from other fungi and bacteria, no plant sequences were identified to show a significant similarity to *As*FMOL. Through the BLAST search against the SRA datasets from plant species of PC1, DNA sequence similarity hits for *As*FMOL were only obtained from *Agrostis* and *Deyeuxia* species, and no corresponding sequence was found from *P. fugax*, harding grass, or oat (Table 1). An eukaryote transcriptome SRA dataset is typically generated through shotgun sequencing of entire mRNA molecules from a target tissues in an unbiased (non-selective) manner, and such a dataset may contain some amount of sequencing artifacts and/or short reads derived from unintended organisms, especially parasitic and pathogenic microorganisms, which could affect the subsequent data annotation [17]. A control BLAST-based search was, therefore, performed using an *Epichloë*-specific sequence, makes caterpillars floppy (mcf)-like gene (GenBank UI: KJ502561.1) [14].

**Table 1. Sequence similarity hit number and highest identity from Pooideae species.**

| Common name | Scientific name | NCBI SRA UI | Tissues | Source | Instrument | Data size (bp) | AsBGNL | | AsFMOL | | Institute/ Organization |
|---|---|---|---|---|---|---|---|---|---|---|---|
| | | | | | | | Highest identity | Hit number | Highest identity | Hit number | |
| Creeping bentgrass | *Agrostis stolonifera* | SRX2962769 | - | Transcriptome | Illumina HiSeq 2000 | 11.5G | 100%, 1e-69 | 921 | 100%, 3e-71 | 102 | Gansu Agricultural University |
| Hair grass | *Agrostis scabra* | SRX2582777-SRX2582785 | Leaf | Transcriptome | Illumina MiSeq | 24.2G | 100%, 2e-153 | 717 | 99%, 4e-152 | 25 | Rutgers University |
| - | *Deyeuxia angustifolia* | SRX692543 | Leaf | Transcriptome | Illumina HiSeq 2000 | 9.3G | 100%, 3e-43 | 4643 | 100%, 3e-44 | 183 | Institute of Natural Resources and Ecology, Heilon |
| - | *Polypogon fugax* | SRX815938 | Entire fresh plant | Transcriptome | Illumina HiSeq 2000 | 4.6G | 98%, 1e-39 | 448 | N.S. | - | Anhui Academy of Agricultural Sciences |
| - | *Polypogon fugax* | SRX815968 | Entire fresh plant | Transcriptome | Illumina HiSeq 2000 | 4.9G | 84%, 2e-54 | 801 | N.S. | - | Anhui Academy of Agricultural Sciences |
| Oat | *Avena sativa* | SRX3481669 | Leaf | Transcriptome | Illumina HiSeq 2500 | 22.6G | N.S. | - | N.S. | - | The Sainsbury Laboratory |
| Oat | *Avena sativa* | SRX3481668 | Leaf | Transcriptome | Illumina HiSeq 2500 | 27.7G | N.S. | - | N.S. | - | The Sainsbury Laboratory |
| Harding grass | *Phalaris aquatica* | SRX669405 | - | Transcriptome | Illumina HiSeq 2000 | 10.2G | N.S. | - | N.S. | - | Teagasc |

'N.S.' stands for no significant hit.

No significantly similar sequence from each SRA dataset was found, indicating absence of an *Epichloë* species-originated sequence in the datasets.

For further validation, the PCR-based screening was performed. The assay with the *As*BGNL-specific primers indicated that the ß-1,6-glucanase-like sequences were only present in creeping bent grass, common bent, and annual beard grass, but not in harding grass (Figs 2A and S3). With the *As*BGNL-specific primers, PCR amplification was not observed from the *Epichloë* gDNA template. Absence of *Epichloë* and *Claviceps* gDNA in the plant gDNA samples was confirmed using the fungus-specific PCR primers. The assay with *As*FMOL-specific primers indicated that the FMO(-like) sequence is present in creeping bent grass and common bent, but not in annual beard grass and harding grass (Fig 2A). Using the *As*FMOL-specific primers, PCR amplicons from *E. festucae* were also observed. The DNA sequence alignment identified SNVs between *As*FMOL and the corresponding *Epichloë* genes, and, in the PCR amplicons, two SNVs were found to be related to an *Aat*II recognition site of *Epichloë* sequence (S2 Fig). Due to these SNVs, the PCR amplicons from *Epichloë* species were predicted to be digested with the *Aat*II restriction enzyme, while the amplicons from creeping bent grass do not possess any *Aat*II recognition site. Through the PCR-RFLP assay, a clear size difference between plant- and fungus-derived DNA fragments was observed, and this result excluded the possibility of presence of *Epichloë* gDNA in plant gDNA samples (Fig 2B).

Using the RT-qPCR-based approach, the gene expression analysis was performed for *As*BGNL and *As*FMOL. The Cq values for *As*BGNL and *As*Act were between 24–28, and the analysis revealed that *As*BGNL was ubiquitously expressed in the young seedlings, including root tissues (Fig 3). This trend was similar to that of *Lp*BGNL in perennial ryegrass [9]. As

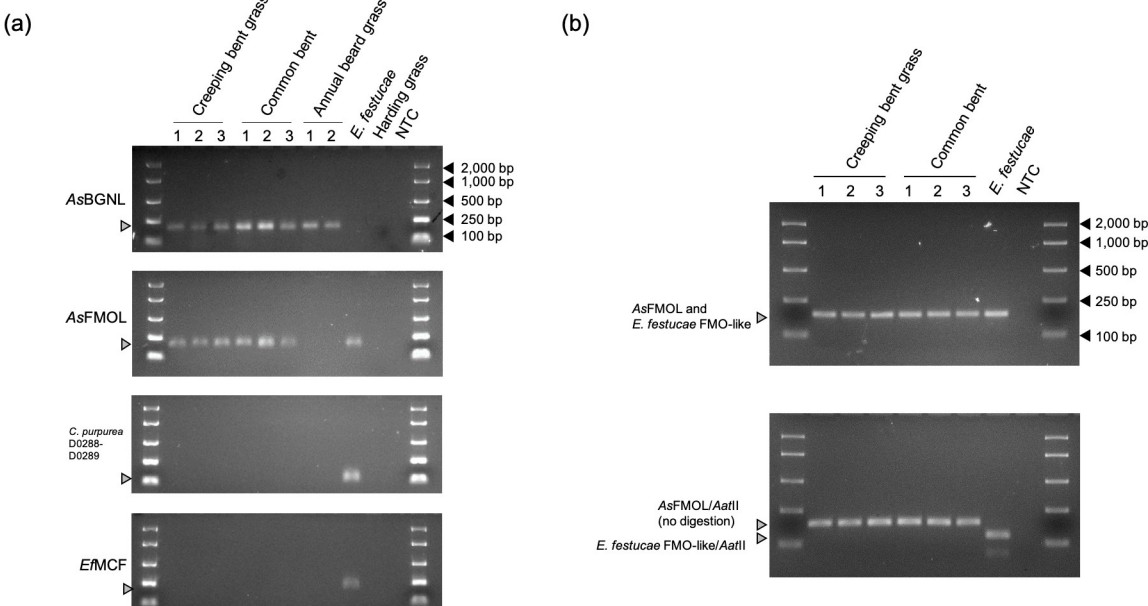

**Fig 2. PCR-based screening for the HGT candidates.** (a) Results of PCR amplification with the AsBGNL/LpBGNL_F and AsBGNL_R primers (top panel), and the AsFMOL_con_f1_AstII and AsFMOL_con_r1_AstII primers (second panel). As a control experiment, PCR with the C.purpurea_D0288F and R primers, and the Epichloe_mcf_F and R primers was performed (third and bottom panels). As PCR templates, gDNA samples from 3 creeping bent grass individuals, 3 common bent individuals, and 2 annual beard grass individuals were used, along with those of *E. festucae* and harding grass. NTC stands for 'no template control'. The expected positions of PCR amplicons on the agarose gel are indicated with grey arrows. The BIOLINE EasyLadder I was used as size standard and the sizes of ladders are indicated with black arrows. (b) Results of the PCR-RFLP assay. The upper panel shows PCR fragments from each gDNA sample before the restriction enzyme treatment, and the lower panel shows the fragments after the treatment.

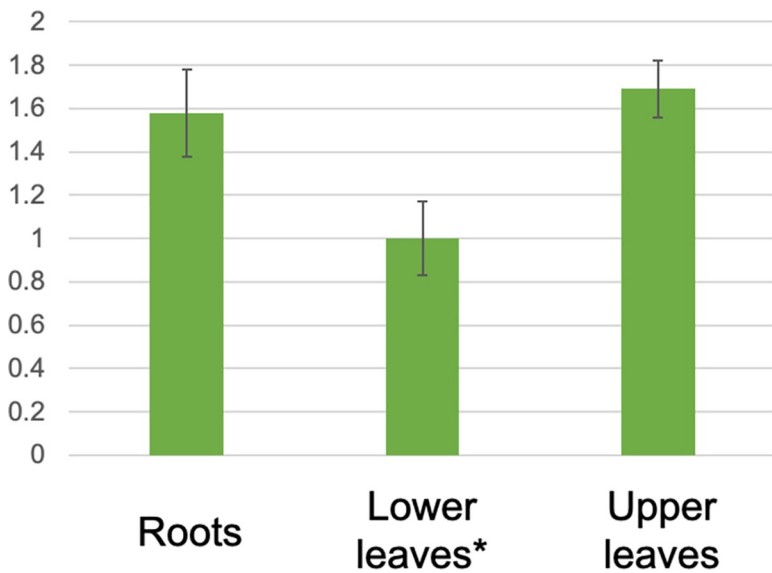

**Fig 3. Gene expression analysis for *As*BGNL.** The expression levels are normalised with that of lower leaves, which is indicated with an asterisk (*). Black lines stand for standard deviation.

*Epichloë* species do not infect root tissues, *AsBGNL* may contribute to plant pathogen resistance, especially against soil-borne fungi, rather than interaction between the fungal endophyte. Although the RT-qPCR assay was performed for *As*FMOL, amplification was detected only after 32 reaction cycles. (S4 Fig). The RT-qPCR data for *As*FMOL were, therefore, concluded not to be suitable for a relative gene expression analysis [18,19]. This assay, however, suggested that the gene was ubiquitously expressed, but at a low level, in young seedlings. From the SRA data of creeping bent grass, hair grass and *Deyeuxia*, a substantially lower expression pattern of *As*FMOL was also suggested, compared with *As*BGNL. The read numbers per kilobase per million (FPKM) of *As*FMOL were only 0.2–1.3, meanwhile those of *As*BGNL were 5.6–31.5 (S5 Table), supporting the result from the RT-qPCR-based assay.

The phylogenetic analysis was performed for *As*BGNL and *As*FMOL. The analysis suggested a closer relationship of *As*BGNL between *Lp*BGNL, than corresponding genes of *Epichloë* species (Fig 4A). The previous study demonstrated the specificity of the plant ß-glucanese-like gene to the Lollinae and Dactylidinae subtribes of PC2, and the phylogeny, thus, suggested a possibility of horizontal transfer of the Lollinae ß-glucanese-like gene into a common ancestor of Agrostidinae species. Other scenarios could be considered, such as two independent (*Epichloë*-Lollinae/Dactylidinae and *Epichloë*-Agrostidinae) HGT events, and HGTs into a common ancestor of the Pooideae species (both PC1 and PC2), followed by lineage-specific gene deletions. However, these alternative scenarios are not supported with the result of the phylogenetic analysis. In case of the plant-to-plant HGT, it is likely that the ß-glucanese-like gene was laterally transferred after the period that broad-leaved Loliinae, including perennial ryegrass, diverged from fine-leaved Loliinae, including sheep fescue (*Festuca ovina* L.). The phylogenetic analysis suggested a closer relationship between *As*FMOL and the corresponding gene of *E. baconii*, *E. amarillans*, *E. mollis*, and *E.festucae*, compared with other *Epichloë* species (Fig 4B). It was, therefore, suggested that the gene was laterally transferred from the *Epichloë* lineage after diversification of the *Epichloë* genus.

*Poa* and sheep fescue are categorised into PC2, and horizontal transfer of the *Poa* cytosolic enzyme phosphoglucose gene, *PgiC*, into sheep fescue has been well characterized [20,21]. Due

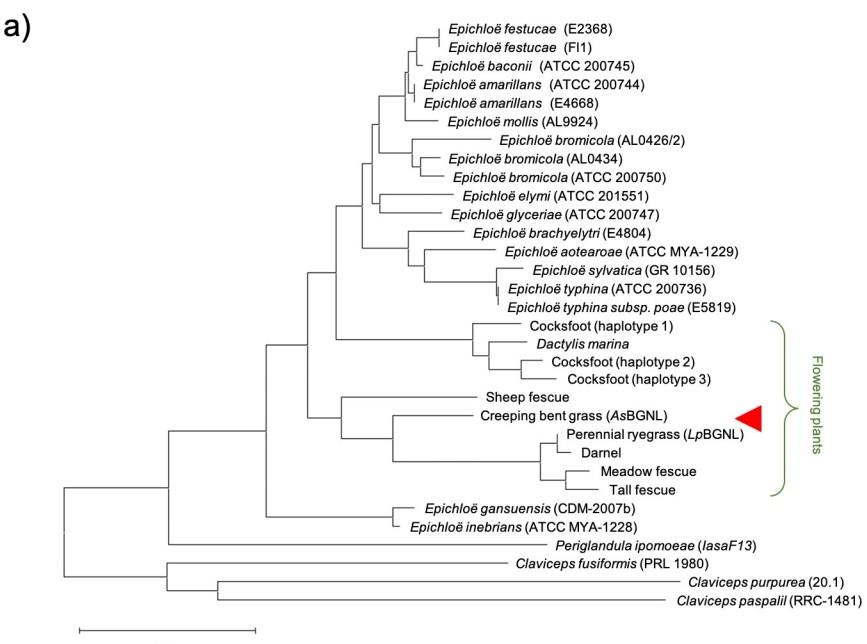

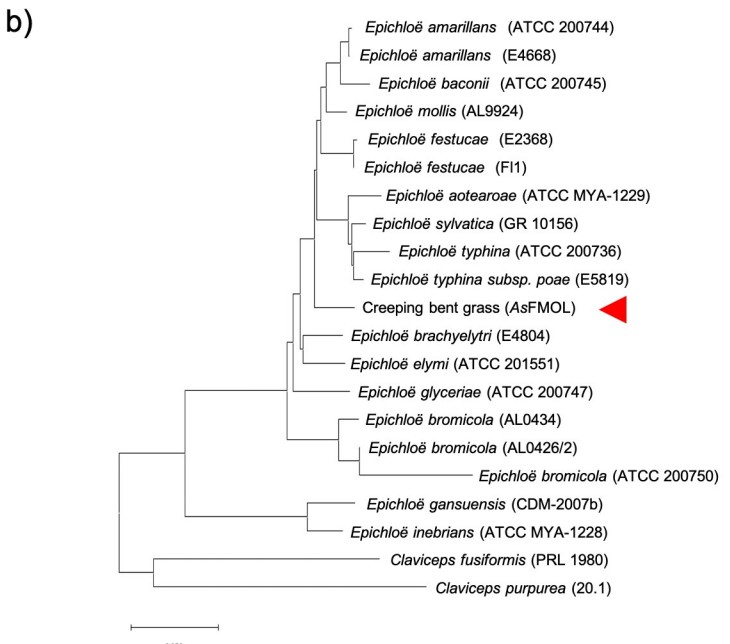

**Fig 4. Phylogenetic analysis for the HGT candidates.** (a) Phylogenetic tree of plant and fungal ß-1,6-glucanase(-like) genes, based on an amino acid alignment of the aryl-phospho-beta-D-glucosidase domain. (b) Phylogenetic tree of plant and fungal FMO-like genes on an amino acid alignment. Sequences from creeping bent grass are indicated with red arrows. Strain identifiers of the Genome Project at the University of Kentucky website or NCBI UI are shown in brackets.

to absence of evidence for cross-fertilisation between the two species, a possibility of microorganism-mediated HGT was initially discussed, rather than fertilisation (introgression)-based gene transfer [20]. Afterwards, a possibility of transformation (gDNA fragment integration)-

like and mRNA-mediated gene exchanges between taxonomically distant plants, especially between parasitic and host species, was discussed [22,23]. The gene exchange between Lollinae and Agrostidinae common ancestors may have occurred through one of the proposed plant-to-plant HGT models. A PCR assay inferred conservation of the intron position between *As*BGNL and *Lp*BGNL (S5 Fig). It is likely that a DNA molecule-mediated mechanism was involved in the gene transfer event of the ß-gulcanese(-like) gene.

Recently, systematic surveys were conducted to identify genes transferred between grass species [24,25]. A total of 26 plant-to-plant HGT candidates were identified from black seed grass (*Alloteropsis semialata*, Panicoideae subfamily), of which 25 were presumably derived from distant Panicoideae linages [24]. The other candidate was concluded to have originated in the Chloridoideae lineage, which is a sister clade of Panicoideae. A lager survey including 17 grass species of the Poaceae family identified 135 plant-to-plant HGT candidates [25]. The survey included 9 Panicoideae species, and a great portion of the candidates were concluded to be genes transferred between Panicoideae lineages. From the Pooideae subfamily, only 3 species [*Brachypodium distachyon*, barley (*Hordeum vulgare* L.), and common wheat (*Triticum aestivum* L.)] were subjected, to find 12 candidates. The current study has suggested that *As*BGNL was acquired through a plant-to-plant HGT event. Although the Poeae tribe includes over 2,500 species, composing the largest grass tribe, no Poeae species were included in the previous survey [5,25]. A further investigation using Poeae species may identify such HGT candidates.

Due to the high amino acid sequence identity to the *Epichloë* FMO, it is possible that *As*FMOL still retain the molecular function. FMO commonly acts on sulfur and nitrogen-containing nucleophiles [26]. This type of monooxygenase has been found in bacteria, animals, fungi and plants, suggesting an ancient origin [26]. Although previous studies in animals and plants indicated that some types of FMOs contribute to longevity, other FMOs may have varied roles, depending on taxonomy. In animals, several FMOs have been reported to be related to metabolic diseases, while those in plants were identified to be related to pathogen defense and auxin biosynthesis [26,27]. A further analysis is essential to reveal the function of *As*FMOL.

In the current report, identification of the novel candidates for horizontally transferred genes in cool-season grasses has been described. Although models for the HGT events are still unclear, the genes possibly retain the molecular functions. The two genes are likely to have transferred across species boundaries relatively recently in the evolutionary time. A further investigation of the two genes may support the hypothesis that HGT has been a part of evolution and adaptation mechanisms of flowering plants (Angiosperms) [24].

## Supporting information

**S1 Fig. DNA sequence alignment of the glucanase(-like) genes.** mRNA sequences from *E. amarillans* and creeping bent grass are aligned with a part of genomic sequence from perennial ryegrass. A dash (-) shows a gap in the DNA sequences, and an asterisk (*) under the alignment denotes 'conserved nucleotide'. The intron sequence of perennial ryegrass glucanase-like gene is shown with an empty box. The location and direction of each PCR primer are shown with a blue arrow.
(PDF)

**S2 Fig. DNA sequence alignment of the flavin-containing monooxygenase(-like) genes.** The *As*FMOL sequence is aligned with *E. amarillans* and *E. festucae* FMO sequences. A dash (-) shows a gap in the DNA sequences, and an asterisk (*) under the alignment denotes 'conserved nucleotide'. The location and direction of each PCR primer are shown with a blue arrow. The *Aat*II restriction enzyme recognition site in the *E. amarillans* and *E. festucae*

sequences is indicated with an empty box.
(PDF)

**S3 Fig. PCR-based screening using the *As*BGNL and *Lp*BGNL locus-specific primers.** The gDNA samples of creeping bent grass, oat, harding grass, perennial ryegrass, and *E. festucae* were used as DNA template. A control experiment with the florigen candidate gene (HD3/FT)-specific primers was performed, to confirm the quality of plant gDNA samples. NTC stands for 'no template control'. The PCR amplicons were visualised on an agarose gel (2% w/v) containing the SYBR Safe stain, and the expected size of PCR amplicons is indicated with a grey-filed arrow. The BIOLINE EasyLadder I was used as size standard.
(PDF)

**S4 Fig. qPCR amplification plot for the *As*FMOL gene expression analysis.** The vertical and horizontal axes indicate the relative fluorescence units (RFU) and the number of PCR cycles. cDNA samples from roots, lower leaves, and upper leaves were used as DNA template. NTC stands for 'no template control'. The sample names are shown on the right side of the plot. The threshold line is shown with the thick green line. The plot was generated with CFX Maestro Software (BioRad).
(PDF)

**S5 Fig. PCR assay with the primers designed across the intron (upper panel) and on the exon/intron boundaries (lower panel) of *As*BGNL.** The gDNA samples of creeping bent grass and common bent grass, and cDNA sample of creeping bent grass were used as DNA template. NTC stands for 'no template control'. PCR amplicons were visualised on an agarose gel containing the SYBR Safe stain. Based on the DNA sequence alignment result (S1 Fig), the AsBGNL_exon_F and R primers were designed. A larger fragment sizes from the gDNA template suggested presence of intron(s) in *As*BGNL (upper panel). The forward primer (AsBGNL_intron_F) was designed across exon/intron boundaries of *Lp*BGNL. The combination of the AsBGNL_intron_F and AsBGNL_R primers amplified DNA fragments from the cDNA templates, but no PCR fragments were observed from the gDNA templates, suggesting conservation of the intron position between *As*BGNL and *Lp*BGNL (lower panel). The expected positions of PCR amplicons from cDNA templates are indicated with the grey-filled arrows. The NEB 100 bp DNA Ladder (upper panel) and BIOLINE EasyLadder I (lower panel) and were used as size standard, and the sizes of representative ladders are shown with the black-filled arrows.
(PDF)

**S1 Table. Plant materials used for PCR-based screening.** SARDI UI denotes the unique identifier of the South Australian Research and Development Institute.
(PDF)

**S2 Table. PCR primers designed and used in the current study.**
(PDF)

**S3 Table. DNA sequence homology search result between creeping bent grass and *E. amarillans* transcriptomes.** The similarity hits for *As*BGNL and *As*FMOL are highlighted with yellow.
(PDF)

**S4 Table. BLAST DNA sequence similarity search using the *As*FMOL sequence as query.**
(PDF)

**S5 Table. Short-read sequencing-based gene expression analysis.** Read count quantification was preformed using the NCBI BLAST tool. The word size parameter was set at 64. The *As*BGNL and *As*FMOL sequences were used as query sequence.
(PDF)

**S1 Raw images.**
(PDF)

## Acknowledgments

The authors would like to thank SARDI for provision of plant seeds.

## Author Contributions

**Conceptualization:** German C. Spangenberg, Hiroshi Shinozuka.

**Formal analysis:** Yugo Watanabe, Hiroshi Shinozuka.

**Funding acquisition:** German C. Spangenberg.

**Investigation:** Yugo Watanabe, Hiroshi Shinozuka.

**Methodology:** Hiroshi Shinozuka.

**Project administration:** German C. Spangenberg.

**Supervision:** German C. Spangenberg.

**Validation:** German C. Spangenberg.

**Writing – original draft:** Yugo Watanabe, German C. Spangenberg, Hiroshi Shinozuka.

**Writing – review & editing:** Yugo Watanabe, German C. Spangenberg, Hiroshi Shinozuka.

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
