## [Decision Letter · Decision Letter 0]

3 Aug 2021

PONE-D-21-20796

Fungus-originated glucanase and monooxygenase genes in creeping bent grass (Agrostis stolonifera L.)

PLOS ONE

Dear Dr. Shinozuka,

Thank you for submitting your manuscript to PLOS ONE. After careful consideration, we feel that it has merit but does not fully meet PLOS ONE’s publication criteria as it currently stands. Therefore, we invite you to submit a revised version of the manuscript that addresses the points raised during the review process.

ACADEMIC EDITOR: As Peer-reviewers suggest modifications in the manuscript address all their comments critically and resubmit your manuscript.

We look forward to receiving your revised manuscript.

Kind regards,

Shunmugiah Veluchamy Ramesh, PhD

Academic Editor

PLOS ONE

Journal Requirements:

Additional Editor Comments:

Reviewers suggest minor revisions to the manuscript. I suggest you to address their comments critically  and resubmit your manuscript for further consideration. Adequate discussion of published research pertaining to horizontal gene transfer is warranted.

Reviewers' comments:

Reviewer's Responses to Questions

**Comments to the Author**

1. Is the manuscript technically sound, and do the data support the conclusions?

Reviewer #1: Yes

Reviewer #2: Yes

Reviewer #3: Yes

2. Has the statistical analysis been performed appropriately and rigorously? 

Reviewer #1: Yes

Reviewer #2: N/A

Reviewer #3: Yes

3. Have the authors made all data underlying the findings in their manuscript fully available?

Reviewer #1: Yes

Reviewer #2: Yes

Reviewer #3: Yes

4. Is the manuscript presented in an intelligible fashion and written in standard English?

Reviewer #1: Yes

Reviewer #2: Yes

Reviewer #3: Yes

5. Review Comments to the Author

Reviewer #1: The authors compared the transcriptome data between bent grass and E. amarillans and identified two horizontally transferred candidate genes. Phylogenetic analysis showed that BGNL was transferred into a common ancestor of Agrostidinae species, while FMOL was transferred into Agrostis stolonifera from Epichloe lineage. The research is structured in an appropriate manner. As there is no functional analysis, the last paragraph "adaptation mechanisms of flowering plants (Angiosperms)" is an over interpretation from the present results. Overall, this manuscript is technically rigorous and meets the scientific and ethical standard, and only minor corrections are needed.

Fig.1a

As Perennial ryegrass is not the main focus of this study, it is better to place it in the third lane.

Furthermore, is there a clear reason why Fig.1a should include the amino acid sequence of perennial ryegrass and Fig.1b should contain the amino acid sequence of E. festucae?

Is there a problem with comparing only E. amarillans and Creeping bent grass?

L241-L244

I could not understand why it is unsuitable for calculating relative gene expression in AsFMOL. Why not just increase the number of cycles?

L269

fine--leaves -> fine-leaves

L307

in cool cool-season grasses -> in cool-season grasses

Reviewer #2: 1. Fungus-originated glucanase and monooxygenase genes in creeping bent grass manuscript is well planned and comprehensive, covering all the aspects of recent studies have revealed presence of fungus-originated genes in genomes of cool-season grasses, suggesting occurrence of multiple ancestral gene transfer events between the two distant lineages. The current article describes identification of glucanase-like and monooxygenase-like genes from creeping bent grass, as lateral gene transfer candidates. The quality of language is good and flow of ideas is easily be followed by the reader. The strength of this manuscript is that discussed almost all the recent research articles related to the research and listed in reference list.

2. The quality of the figures and/or table is satisfactory

3. The manuscript covered the topic in an objective and analytical manner

and reference list covered the relevant literature adequately and in an unbiased manner.

4. I would suggest this manuscript is accepted only after minor corrections are incorporated as suggested below.

Suggested corrections:

In Text:

Some references are not according to the journal format, it has to be modified accordingly.

Reviewer #3: In this research glucanase-like and monooxygenase-like genes were identified as lateral gene transfer candidates in creeping bent grass. In- silico studies were conducted using transcriptome datasets to identify these genes in related plant species. PCR based screening were further performed to confirm the results. In addition gene expression studies were conducted to validate the expressions of these genes. Overall this research provides important information regarding the later genes transfer in creeping bent grass and the scientific quality of the research is high. Following point may be considered.

There are previously published reports regarding the lateral gene transfer in grasses. Authors should discuss in more details regarding the major outcome of this research and highlight its importance.

6. PLOS authors have the option to publish the peer review history of their article (what does this mean?). If published, this will include your full peer review and any attached files.

Reviewer #1: **Yes: **Ryo Fujimoto

Reviewer #2: No

Reviewer #3: No

---

## [Author Response · Author response to Decision Letter 0]

18 Aug 2021

Dear Editor,

We would like to appreciate the Reviewers for their generous comments on the manuscript. The followings are point-to-point responses to the Reviewers.

Comment 1 from Reviewer #1: As there is no functional analysis, the last paragraph "adaptation mechanisms of flowering plants (Angiosperms)" is an over interpretation from the present results. 

Response for comment 1: The sentence was corrected following the comment (Lines 329-330 of the marked-up copy). How horizontally transferred genes contributed to recipient’s adaptation is one of current argument in this field. A citation was added (Dunning et al. 2019 doi:10.1073/pnas.1810031116).

Comment 2 from Reviewer #1: Fig.1a As Perennial ryegrass is not the main focus of this study, it is better to place it in the third lane.Furthermore, is there a clear reason why Fig.1a should include the amino acid sequence of perennial ryegrass and Fig.1b should contain the amino acid sequence of E. festucae? Is there a problem with comparing only E. amarillans and Creeping bent grass?

Response for comment 2: The alignment was from the CLUSTALW program, and no manual adjustment was performed. The reason why perennial ryegrass sequence is included in the AsBGNL figure is that the perennial ryegrass sequence showed a higher sequence identity with AsBGNL than the E. amarillans sequence did. Regarding the AsFMOL figure, E. festucae is a representative species of Epichloe and the genome of E. festucae has been better assembled than other Epichloe species in the database (Schardl et al. 2013 doi:10.1371/journal.pgen.1003323). As Epichloe genomes in the database have not been completely assembled, we believe that inclusion of multiple data points is essential, to eliminate a possibility of sequencing contamination or assembly artifacts. 

Comment 3 from Reviewer #1: L241-L244 I could not understand why it is unsuitable for calculating relative gene expression in AsFMOL. Why not just increase the number of cycles?

Response for comment 3: A common question for an HGT candidate is whether the gene is expressed in the recipient species. The main purpose of the experiment is to determine this. In a qPCR assay, higher Cq values are generally less reliable, due to generation of some amounts of PCR artifacts (primer dimers and non-specific product) and a higher sample variance (Ruiz-Villalba et al. 2021 doi:10.3390/life11060496; Taylor et al. 2019 doi:10.1016/j.tibtech.2018.12.002). Therefore, the cutoff value should be set to somewhere between 30-35 cycles. In our case, the Cq values from more than half samples were over 35 cycles, to conclude that the qPCR result was not suitable for the relative expression analysis. We also tried other PCR primer combinations, to find that none of them worked better than the AsFMOL_ps_f1 and r1 combination. The references for the qPCR QC process have been added in the text (Line 243 of the marked-up copy). 

Comment 4 from Reviewer #1: L269 fine--leaves -> fine-leaves, L307 in cool cool-season grasses -> in cool-season grasses

Response for comment 4: The text was corrected following the comments.

Comment from Reviewer #2: Some references are not according to the journal format, it has to be modified accordingly.

Response for comment: The reference list was generated the Zotero program (https://www.zotero.org). Then, the list was manually corrected, which can be found in Lines 346-436 of the marked-up copy.

Comment from Reviewer #3: There are previously published reports regarding the lateral gene transfer in grasses. Authors should discuss in more details regarding the major outcome of this research and highlight its importance.

Response for comment: Discussion about the recent progress in grass species, especially in Panicoideae subfamily was inserted, which can be found Lines 296-311 of the marked-up copy. For this section, two references (Dunning et al. 2019 doi:10.1073/pnas.1810031116; Hibdige et al. 2021 doi:10.1111/nph.17328) were added

I believe that all points raised by the Reviewers have been addressed.

Hiroshi Shinozuka

On behalf of all authors.

---

## [Decision Letter · Decision Letter 1]

25 Aug 2021

Fungus-originated glucanase and monooxygenase genes in creeping bent grass (Agrostis stolonifera L.)

PONE-D-21-20796R1

Dear Dr. Shinozuka,

We’re pleased to inform you that your manuscript has been judged scientifically suitable for publication and will be formally accepted for publication once it meets all outstanding technical requirements.

Kind regards,

S.V. Ramesh, PhD

Academic Editor

PLOS ONE

Additional Editor Comments (optional):

Reviewers' comments:

Reviewer's Responses to Questions

**Comments to the Author**

1. If the authors have adequately addressed your comments raised in a previous round of review and you feel that this manuscript is now acceptable for publication, you may indicate that here to bypass the “Comments to the Author” section, enter your conflict of interest statement in the “Confidential to Editor” section, and submit your "Accept" recommendation.

Reviewer #1: All comments have been addressed

2. Is the manuscript technically sound, and do the data support the conclusions?

Reviewer #1: Yes

3. Has the statistical analysis been performed appropriately and rigorously? 

Reviewer #1: Yes

4. Have the authors made all data underlying the findings in their manuscript fully available?

Reviewer #1: Yes

5. Is the manuscript presented in an intelligible fashion and written in standard English?

Reviewer #1: Yes

6. Review Comments to the Author

Reviewer #1: The authors responded my comments and I agreed with their comments. Thus I do not have further comments.

7. PLOS authors have the option to publish the peer review history of their article (what does this mean?). If published, this will include your full peer review and any attached files.

Reviewer #1: No

---

## [Editor Report · Acceptance letter]

2 Sep 2021

PONE-D-21-20796R1 

Fungus-originated glucanase and monooxygenase genes in creeping bent grass *(Agrostis stolonifera L.)*

Dear Dr. Shinozuka:

I'm pleased to inform you that your manuscript has been deemed suitable for publication in PLOS ONE. Congratulations! Your manuscript is now with our production department. 

Kind regards, 

on behalf of

Dr. Shunmugiah Veluchamy Ramesh 

Academic Editor

PLOS ONE